# Priority Determination to Apply Artificial Intelligence Technology in Military Intelligence Areas

**Sungrim Cho [1], Woochang Shin [2], Neunghoe Kim [3], Jongwook Jeong [4]**  **and Hoh Peter In [3],***

[1] Korea Institute for Defense Analyses and Department of Computer Science and Engineering, Korea University, Seoul 02841, Korea; srcho@korea.ac.kr
[2] Department of Computer Science, SeoKyeong University, Seoul 02713, Korea; wcshin@skuniv.ac.kr
[3] Department of Computer Science and Engineering, Korea University, Seoul 02841, Korea; nunghoi@korea.ac.kr
[4] Department of Computer Science, Korea Military Academy, Seoul 01805, Korea; jw385@mnd.go.kr
* Correspondence: hoh_in@korea.ac.kr

**Abstract:** The Ministry of National Defense of South Korea is currently acquiring various surveillance and reconnaissance assets to improve its independent surveillance and reconnaissance capabilities. With the deployment of new strategic and tactical surveillance and reconnaissance assets, the amount of information collected will increase significantly, and military intelligence capable of handling greater complexity will be needed to process such information. As a consequence, it will no longer be possible to handle the increased workload through a manual analysis conducted by intelligence analysists. Further, the number of intelligence analysists is expected to decrease in the near future owing to a reduction in the total number of troops, thereby exacerbating the need to apply artificial intelligence technology to process military intelligence tasks more quickly and accurately. In this study, a method is introduced for determining the ways to prioritize the AI technology domains applied to military intelligence. Consequently, among the five stages used, the processing stage has the highest priority. The application of AI technology to all the stages of information circulation may be ideal. Nevertheless, among various military intelligence domains, the one that affords the highest effectiveness of such an application should be prioritized. This is owing to resource and defense budget limitations.

**Keywords:** priority determination; artificial intelligence; military intelligence; surveillance; reconnaissance

## 1. Introduction

The Ministry of National Defense (MND) of South Korea is striving to acquire various surveillance weapon systems, such as military satellites, high/medium-altitude unmanned aerial vehicles, and drones, to improve real-time battlefield-situation awareness. As new strategic and tactical surveillance assets are being deployed, the amount of information collected will significantly increase, and the military intelligence required to deal with them will also become complicated. However, the manpower-centered way of analyzing the collected intelligence based on the experience of intelligence analysts will not only fail to handle the growing information, but also reduce the number of people who can serve as intelligence analysts in the near future, owing to a reduction in the number of troops [1].

Therefore, the need to apply the artificial intelligence (AI) technology to conduct quick and accurate military intelligence work will be increased [2]. Military intelligence services comprise the following five stages of information circulation: planning, collection, processing, analysis, and dissemination [3].

Notably, the intelligence analysts use information systems at some stages but manually perform most of the tasks. Although it may be ideal to apply the AI technology to all the stages of information circulation, we must prioritize on which domain of military intelligence areas it is most effective to apply the technology while considering limited resources and budgets.

This paper is structured as follows. In Section 2, we discuss the concept and work status of military intelligence, trends in the development of military information systems, and national strategies for AI. In Section 3, we introduce a priority determination method that comprises target selection, priority determination element, priority determination procedure, among others, for the application of AI. In Section 4, we verify the applicability and effectiveness of the proposed method by using the priority determination method for applying AI to military intelligence areas. In the last chapter, we present the summary, implications, and future research directions of this study.

## 2. Research Background and Trend Analysis

### 2.1. Concepts and Tasks of Military Intelligence

John Boyd proposed the observe-orient-decide-act (OODA) loop as a combat decision-making model in his book "A discourse on winning and losing" published in 1987 [4]. The OODA loop forms a circular loop connected by the observation–orient–decide–action phase. If we are to preemptively attack and neutralize the enemy, we must precede OODA loop faster than the enemy. The first two stages of the OODA loop, observation and orient, are related to military intelligence. To speed up the operation tempo, we must reduce the time of these two stages.

Military intelligence or intelligence refers to the useful products created by collecting, processing, using, and analyzing the data associated with enemy, potential enemy, foreign, and other operational environment required for mission performance. The term "information" and "intelligence" is used separately. Information refers to the data collected to produce intelligence and processed in a form that is understandable to users but not analyzed. Intelligence is applied to visualize battlefield situations, grasps enemy intentions and capabilities, identifies enemy centers, and supports accurate future predictions to successfully perform military operations. It is classified into geospatial intelligence, weather intelligence, human intelligence, signal intelligence, measurement and symbol intelligence, open-source intelligence, cyber space intelligence, technical intelligence and media intelligence depending on the source of the collection, such as human, document, and machinery.

Military intelligence services refer to a series of intelligence activities to collect information according to the priority information request by the commander and then disseminate the intelligence to the commanders/staff and related departments so that the intelligence collected can be used for military operations. Military intelligence services are performed in the five stages of planning, collection, processing, exploitation, and dissemination [5].

### 2.2. Development Trends of Military Information Systems

Various military information systems were developed and operated to support military intelligence services. Analyzing the development trend of military information systems, the following characteristics can be seen. First, an information system was developed to support one stage, such as planning and processing, in the information circulation stage. However, recently, the scope of development has been expanded to all the stages of information circulation, ranging from the planning to dissemination stage [6,7].

Second, the information collected using a single sensor was analyzed, but it is being developed to increase the accuracy of information analysis by fusing the information collected using multiple sensors [8]. In other words, intelligence analysts are attempting to fuse imagery information collected using sensors, such as satellites, Hawks, and Geumgang, or attempting to fuse imagery, signals, and open-source information (US DCGS, All-Source Analysis System; Raytheon Corporation FoxTEN, 2019 [9]; Rafael IMILITE, 2017 [10]; Jane's Data Analytics, 2019 [11]; Korea ETRI, 2019 [12–15]).

Third, the intelligence analysis was previously centered on a "collection sensor" that analyzed the information collected using surveillance and reconnaissance assets (sensors); however, the analysis now analyzes the information based on the "mission" [16]. The Ministry of National Defense, Israel, is striving to replace IAI ELTA's RICENT by Rafael's IMILITE. Notably, RICent is a "sensor-centered" system that analyzes and disseminates the imagery information collected using sensors. IMILITE is a "mission-centered" system that analyzes and disseminates the imagery information collected by requesting for the images necessary for performing missions [10].

Finally, the amount of the military intelligence data collected owing to the emergence of new surveillance weapons systems, such as military satellites, drones, and unmanned aerial systems, is exponentially growing. In addition, the AI technology is being applied to the analysis work to efficiently process and fuse such data to produce reliable intelligence [17–20]. A typical example is Project Maven. DoD established an Algorithmic Warfare Cross-Functional Team, called Project Maven, which have conducted projects by applying AI to help a workforce that was significantly overwhelmed because of the incoming data, which included millions of hours of video [21].

According to a report released recently released by the RAND corporation [22], there are several advantages in the application of AI in the military domains: improved decision-making speed; effective utilization of big data, targeting, and vision; mitigation of military issues; improved cyber defense; and reduced labor and expenses. Intelligence surveillance and reconnaissance (ISR) corresponds to one of the most recent investments in military AI, and this trend is expected to continue. In particular, considering the amount, speed, and diversity of data collected from sensors on the ground and in air, space, and cyberspace, the report concluded that the partial or complete utilization of AI in a military analysis is necessary.

## 2.3. National Strategy for AI

The United States issued an executive order for an AI initiative in February 2019 [23,24]. The executive order prioritized long-term, proactive investment by the government in AI research, development, and manpower to enhance the competitiveness of the private sector. In particular, the initiative focused on the implementation of AI technology in the fields of next-generation research and development (R&D) and military security, which are difficult to promote through the private companies.

In addition, the Chinese government released the "New Generation Artificial Intelligence Development Plan" in July 2017 [24]. This plan was initiated by the government to foster large-scale investments and increase manpower in the field of AI. The plan is also aimed at designating leading companies for AI development and fostering industry-specific platforms.

The French government also announced its AI strategy in March 2018 [24,25], espousing the adoption of AI, the core of the future digital economy, to solve social problems, and promoted industrialization in the strategic field to solve varying problems regarding occupation, employment, and ethics.

Japan's response to the fourth industrial revolution was based on the Japan Revitalization Strategy in 2015 [26], applied at the joint level of ministries. This strategy emphasized the challenges faced by AI R&D in the context of the technological characteristics of such intelligence and the strengths of the robotics technologies possessed by Japan. Moreover, Japan announced its AI strategy in March 2019 [24,26], which aimed to boost industrial vitality and accelerate AI technology innovation to solve social problems such as low growth and an aging society.

In December 2019, the government announced the "National Strategy for Artificial Intelligence in the Republic of Korea [24]." This strategy, with the vision of "beyond the IT powerhouse, to become the AI powerhouse," includes three areas, nine strategies, and 100 implementation tasks. AI is expected to spread throughout the industry to all domains. In particular, the MND has identified the development of intelligent platforms in 2020 and the intelligence functions of command control systems using defense data as the major task. In addition, it has decided to accelerate the development of intelligence supporting the command system through the establishment of the Defense Intelligence Data Center from the beginning of 2020.

The Ministry of National Defense announced the "4th Industrial Revolution Smart Defense Innovation Plan" in conjunction with "Defense Reform 2.0" by utilizing advanced technologies from the 4th Industrial Revolution in anticipation of all-out threats, including those from North Korea [27]. The plan was envisioned as "building a smart and strong military through defense innovation based on the 4th revolution," and the three major innovation areas of defense operation, technology and infrastructure, and military strength were presented. In the domain of innovation of the military strength, seven major areas and 30 weapon systems projects were selected to spearhead the R&D of advanced defense science and technology. Among them, various weapons systems are related to military intelligence, such as military satellite, unmanned surveillance and reconnaissance aircraft, reconnaissance drones, and multi-source video convergence systems in the R&D core technology area [28].

In addition, the MND has devised its development plans for various military information systems, including a ground force intelligent video convergence sharing system, cloud-based military intelligence dissemination system, and image information management system through the application of AI technologies [29].

## 3. Method of Priority Determination for the Application of AI

To fairly and systematically determine the priority for applying the AI technology to the domain of military intelligence, the following steps are performed: (a) select the target area for applying the technology; (b) prepare the criteria for determining the priority; and (c) establish procedures for determining the priority criteria.

### 3.1. Selection of Target Areas Subject to Technology Application

In this study, the military intelligence service is selected as a subject of technical application. The segmentation of the business domain is critical to determining in which detailed domain the AI should be applied in the military intelligence areas. In this study, the scope of military intelligence services is determined to include all the five stages of "planning, collection, processing, analysis, and dissemination", according to the training manual of the Joint Chiefs of Staff [3].

### 3.2. Criteria for Determining Priorities

The first factor to consider while determining the priority to apply AI technology will be the problem, or requirements, to be solved by applying the AI technology. Next, we must identify which artificial intelligence technology is needed to solve the problem. AI has various sub-sectors, and the speed of technology development in each sector is different. Therefore, the effect of applying the AI technology should also be considered. The availability of data is another major consideration because the AI technology must have sufficient data to learn and apply the data. In this section, we present the criteria for prioritizing the following four factors to apply the AI technology to military intelligence areas.

① Requirements:

First, the requirements should be addressed at the work level, and the difficulty of the requirements should also be considered, such as how clear or how complicated they are.

② Data:

The next factor to consider is the readiness of the data, such as whether the data are available to be applied to the AI technology (data availability), whether the data are machine readable (data readability), and whether the data can be shared (data confidentiality).

③ AI technology:

A classification system for AI technology has not yet been established, and various studies are currently being conducted on this topic. In this study, the classification criteria proposed by the Ministry of Science and ICT, presented in Table 1, have been applied [30]. Examples applications of AI in military operations include machine learning applied to natural language processing and

understanding, the provisioning of information to support the commander's decision making, automatic target detection, automatic target recognition (ATR), and combat robot development. Moreover, AI technologies are required for the selection of specific technologies by considering the data requirements of the application domain.

**Table 1.** Classification of the artificial intelligence (AI) technology.

| Type | | Description |
| --- | --- | --- |
| Learning | Machine Learning | Forms a recognition or understanding model based on data, or analyzes and learns data on its own to find the optimal answers. |
| Inference | Inference/Expression of knowledge | To derive the answer to problems (new information) by itself based on input and learning data |
| Recognition | Visual Intelligence Linguistic Intelligence Auditory Intelligence | To obtain solutions using method equivalent to the sensory organs of human beings, such as looking, reading, and listening |
| Application | Understanding Situation/Emotions | To understand situations and emotions by analyzing sensor data or user data |
| | Intelligent Agent | To assist machine and robot movements as well as human behaviors and judgment to execute learning and judgment results |

Source: The Ministry of Science and ICT [30] p. 2. The description of the table was added.

④ Technical effects:

Various artificial intelligence technologies can be applied to solve the requirements of military intelligence areas. Depending on the nature and skill of the work, the effect of improving the work may be different with AI. Therefore, to determine the priority of each sector, we must consider the effect of improving business while applying the AI technology. Table 2 summarizes the four criteria for determining the priorities described above.

**Table 2.** Criteria for determining the priorities.

| Item | Considerations | Description |
| --- | --- | --- |
| Requirement (Problem) | Clarity of requirements | Is there a problem to solve defined by applying AI? (current or future ISR work) |
| | Complexity of requirements | How complex are the business procedures that need to be handled? (the number of information sources: one/multi-sources) |
| Data | Data availability | Are there digitized data to be applied using the new technology? |
| | Data readiness | Is the data machine readable? |
| | Data confidentiality | Can the data be released and shared? |
| Artificial Intelligence Technology | Maturity level of the AI Technology | Is there an AI algorithm to be applied? Is the technology maturity of the algorithm applicable? |
| Application Effect | Technology Application Effect | How effective is the application of the AI technology? (refer to Project Maven) |

### 3.3. Priority-Determination Procedure

The priority-determination procedure for applying AI is depicted in Figure 1.

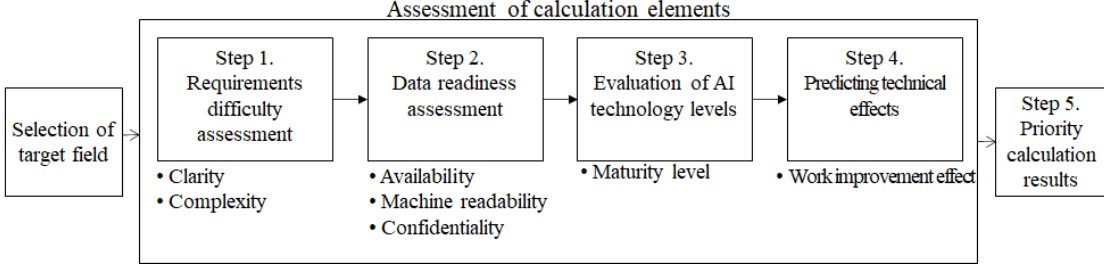

**Figure 1.** Procedure for applying AI.

The first Step is to assess the difficulty of requirements. The difficulty of the requirements is measured on the basis of the clarity and complexity of the requirements. In this study, the clarity of the requirements is high if they are defined as the problems to be addressed in the current work, and the clarity is low if they are not defined as problems that arise from the acquisition of a new surveillance weapon system in the future. The clearer and simpler are the requirements, the easier it becomes to apply new technologies.

For example, one can manually establish an operational plan for the current surveillance and reconnaissance asset during the planning stage, but one must establish an operational plan for the future surveillance and reconnaissance assets by considering various conditions, such as mission, weapon system and sensor characteristics, and weather, among others. However, these problems must be solved in the future, as the requirements are not clearly defined at this time.

Complexity, along with the clarity of requirements, must be considered. If the requirements to be solved by applying the AI technology are simple problems, they are considered less complex. However, if the requirements to simultaneously solve several problems are difficult, then they are considered more complex. The AI technology is easily applied when the complexity of the requirements is low.

In Step 2, the data readiness is assessed. AI technologies require sufficient data for learning. In addition, data are needed for the application and training of AI algorithms. The requisite amount of data is available because of the constant accumulation of current surveillance and reconnaissance assets. However, future data are unavailable because of the nature of the data sources.

Owing to the nature of military intelligence, some data may be unsuitable for use during the application of AI technology because of poor quality. Images of weapon systems are generally difficult to obtain because they reflect the weapon characteristics of a specific country. In addition, because several images depict weapon systems that are covered or shielded to protect them from enemy attack, training an algorithm on such a basis or applying it toward the automatic detection and recognition of a target is difficult to achieve. Furthermore, it may be difficult to employ satellite images because they may be negatively affected by the weather. In these cases, it is necessary to preprocess the data before applying AI technology. For example, improving the resolution of the data can improve the performance of an AI algorithm.

It should be noted that the data must be processed in a machine-readable form. The classification of machine-readable forms is based on the five-level criteria proposed by the Ministry of the Interior and Safety, presented in Table 3 [31].

The first level is an unsatisfied format, including PDFs that cannot be modified or converted into data and can only be read by specific software. The second level is a format that can be read, modified, and converted by specific software to a minimally satisfying format, such as HWP, XLS, and MPEG. Levels 3–5 are open formats that can be read, modified, and converted by any software. Finally, the data confidentiality indicates the confidentiality level of the data. The higher the confidentiality level of military intelligence, the more difficult it is to apply AI technology because data sharing and fusion may be restricted.

**Table 3.** Classification of the data format.

| Level | Machine Readable Form | Characteristics | Example |
|:---:|:---:|:---:|:---:|
| 1 | Unsatisfied format | only specific software (SW) can read but cannot modify and convert | PDF |
| 2 | Minimum satisfying format | only specific SW can read, modify, and convert | HWP, XLS, JPG, MPEG |
| 3 | | all SW can read, modify, and convert | CSV, JSON, XML |
| 4 | Open format | data structure that describes data characteristic relationships based on URIs to be linked and shared with other data on the web | RDF |
| 5 | | | LOD |

Source: The Ministry of the Interior and Safety [31] p. 6.

In Step 3, AI technology level is evaluated. Based on the classification system introduced in Section 3.2 (AI Technology), the technology level of each detailed AI technology is considered. The level of development of the AI technology is different for each detailed technology, and some technologies may show accuracy beyond the human-skill level.

General Adversary Networks (GAN) technology proposed by Ian Goodfellow can create images that are very similar to reality in environments where there is a lack of learning data [32]. SK telecom proposed "Discover Cross-Domain Relations with Generative Adversarial Networks (DiscoGAN)" that transforms data from one domain into data from another domain by discovering the relationship between two different domains [33]. The technology creates and recommends a shoe design or pattern to match a handbag. For example, this technology using machine learning creates or recommends shoe designs or patterns to match a handbag which the user chooses. Naver proposed "StarGAN" that was a deep learning-based image conversion technology that could learn images and change various conditions, such as hair color, gender, and facial expressions of a person in the image at the same time [34].

In the case of the inference question and answer (Q&A) test, DeepMind using Relation Networks (RNs) recognized videos, images, and text and inferred the relationship of them. By applying this algorithm to the CLEVR data set [35], the level of accuracy was 95.5%, which exceeds the level of accuracy of 92.6% for humans [36,37].

In addition, the visual intelligence surpassed the human level of 94.9% with the object recognition accuracy of 96.43% at ImageNet Large Scale Visual Recognition Competition (ILSVRC [38]) in 2015. The accuracy of face recognition has surpassed that of human eyes, and the highest rate of face recognition has reached 99.15% [39]. The deep-learning system of Google achieved an accuracy of 70% to diagnose prostate cancer, which outperformed an average accuracy of 61% by U.S. pathologists [40].

The automatic speech recognition technology (ASR) of Google has word error rates (WER [41]) of 8.5% in July 2016 [42] and 4.9% in 2017 [43,44]. Furthermore, Microsoft yielded a 5.1% WER on the National Institute of Standard and Technology (NIST) 2000 Switchboard test set in 2017 [45]. In 2017, IBM announced that its WER was 5.5 percent using their Long Short-Term Memory (LSTM) and WaveNet language models [46]. Hearing and linguistic intelligence achieved 4.21 points out of 5 points in 2016, which is similar to the human voice level of 4.55 points [47].

The application technology of AI that understands the external characteristics of objects appearing in images and videos is developing. AI application technology is being developed to analyze the correlation of human eyes, nose, and mouth shapes to understand facial expressions or guess emotions [48]. Furthermore, the technology of understanding the situation in the image and expressing it in language is developing [49].

In Step 4, the effect of application of the technology was predicted. Such an application ought to be adjudged in terms of its effectiveness, which indicates the extent to which it improves the performance

of the task at hand. The U.S. intelligence agencies have been conducting on the project Maven to apply AI technology to the information collected using unmanned aerial surveillance vehicles [50]. They predicted the effects of applying the technology in terms of the degree of automation achieved as a result of the project. The technology of converting voice into text has been shown to exhibit a five-fold effect, whereas computer vision and AI technology exhibit a 50-fold effect. Notably, even during the application of AI technologies, the effects of the applications should be considered because they produce different degrees of improvement in the task involved.

In Step 5, the priority for the application of AI is determined by synthesizing the prioritization elements described earlier. Priority-determination elements are expressed as sub-attributes in the two-dimensional graph depicted in Figure 2; the value of the attributes is designed to increase from left to right and from bottom to top. To determine the priority, the score of 1 was given only in the case of (+) in the quadrant. The highest priority is evaluated in the field with the highest score by adding the given scores.

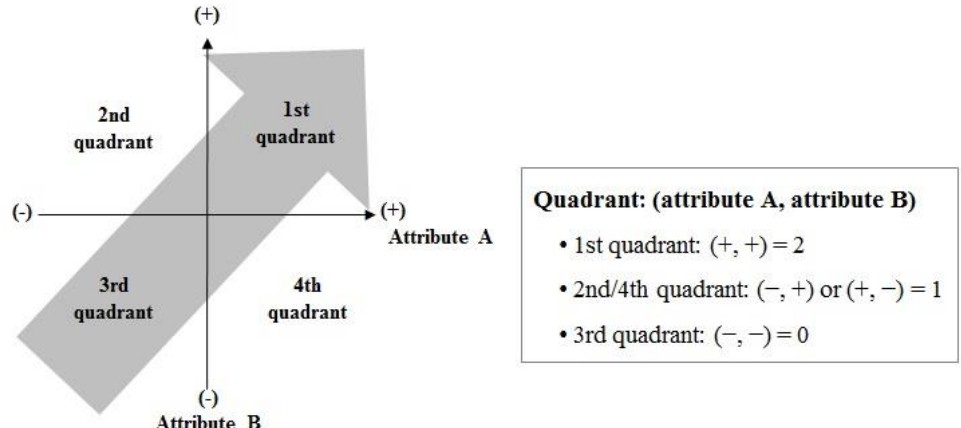

**Figure 2.** Priority determination graph.

The priority of the application of AI technology is calculated using the Equation (1) given below.

$$P_{stage\_i} = C_r \times R_{stage\_i} + C_d \times D_{stage\_i} + C_{Tm} \times TM_{stage\_i} + C_{Te} \times TE_{stage\_i} \tag{1}$$

where $P$ represents the priority of applying AI in the military intelligence area. *Stage_i* points to one of the five stages of planning, collection, processing, analysis, and dissemination. The difficulty of the requirement is represented by $R$, data readiness as $D$, the technology maturity level as $Tm$, and the technology application effect as $Te$. $C$ represents the weight of the evaluation element.

## 4. Applying Priority Determination Methods for Applying AI to Military Intelligence Areas

We demonstrate the case in which the priorities for applying AI to military intelligence areas are determined using the method suggested in Section 3. As previously explained, because the speed of the development of AI technology differs in each sub-sector, the priority should be determined considering both the characteristics of military intelligence areas and AI technology. The priority determination procedure presented in this study provides a criterion for determining the priority regarding which sector of the military intelligence areas we should apply the AI technology to in order to effectively execute a limited defense budget.

### 4.1. Selection of the Target Domain

The military intelligence services comprise five stages of the information circulation, namely, planning, collection, processing, analysis, and dissemination. We analyze the work characteristics of each stage of military intelligence areas to be applied with the AI technology. Before determining

the priorities for applying the AI technology, we consider the characteristics of each information circulation stage.

The planning stage establishes a plan to collect information using surveillance and reconnaissance assets. However, due to the low number of surveillance and reconnaissance assets available in the military, the military intelligence officer coordinates their operation plan using MS Excel. When further surveillance and reconnaissance assets, such as military satellites, are acquired in the future, it would be necessary to establish a collection plan that reflects various conditions, such as missions, types of collection systems and sensors, and weather. This is expected to complicate the collection-planning task, and the development of an information system will be required for a smooth execution of the task.

In the collection stage, information is collected using surveillance and reconnaissance assets. To collect information, there are various surveillance and reconnaissance assets, such as imagery, signal, and open source information. When most surveillance and reconnaissance assets were acquired, the format of the collected information was not standardized. Therefore, commercial satellites produce images in the Geo-TiFF format, HUAV in the NITF format, and MUAV in the JPEG2000 format, and it is difficult to analyze and fuse such non-standardized data.

The processing stage is to read and interpret individual information, such as images, signals, and machinery information, collected using surveillance and reconnaissance assets. In the analysis stage, all-source analysis synthesizes and analyzes the analysis reports that were prepared in the previous processing stage. In the processing stage, the collection troops analyze the information that was collected using a single source and load the reports into the information systems, while the higher-level organization, i.e., Joint Chiefs of Staff, analyzes the information collected using multiple sources or all sources in the analysis stage. In the dissemination stage, the analysis reports are provided to users. The users log into the information system and search for the intelligence reports on bulletin boards on the basis of the information type.

After examining the current status of the intelligence work of planning, collection, processing, analysis, and dissemination, we found out that task managers use Excel and military information systems. However, information systems are being used as simple storage and shared spaces. For example, the information analyst manually analyzes the collected imagery information and then creates and uploads the analysis reports in Hangul [51] or a PowerPoint file, following which he/she obtains the necessary information by searching for keywords in the title of the report on the information bulletin board.

Although the amount of information collected would rapidly increase with the acquisition of various strategic and tactical surveillance and reconnaissance weapons systems in the future, the number of intelligence analysts might decrease because of the reduction in the number of troops. In addition, we must efficiently perform military intelligence areas by applying the AI technology in accordance with the development of the latest information and communication technology.

### 4.2. Priority Determination

To apply the AI technology to military intelligence areas, the procedure presented in Figure 2 was implemented, and the priority was determined. In the first stage, five researchers determined the priorities, and in the second stage, 19 military intelligence officers and IT experts reviewed the determination results during three workshops. The researchers undertook research in the fields of defense informatization and computer science. One researcher had more than 30 years of experience, two had more than 20 years, and two people more than 10 years. The military intelligence officers were in charge of the policy department and the analysis departments of the Joint Chiefs of Staff, the collection and the command and control military units, and the system development organization (the Defense Acquisition Program Administration). On average, they had approximately 20 years of experience.

Priorities are shown in the graph when applying the determination elements of AI technology to the detailed stages of the military intelligence areas.

In Step 1, we evaluate the difficulty of the requirements for each domain in the areas of military intelligence. The clarity of the requirements is a criterion for a successful IT project [52–55]. Many IT projects undertaken in the military have been negatively affected by frequent changes in their requirements. In particular, the exact definition of the requirements is important when developing information systems with the aid of AI technology. The requirements are related to the characteristics of the particular AI technology. This is because the AI algorithm that can be applied varies depending on the requirements.

In the graph depicting the difficulty of requirements, the X-axis indicates the clarity of the requirements, which increases when moving toward the right, and the Y-axis represents the complexity of the requirements, and the requirements become simpler toward the vertical direction. The clearer and the simpler the problem is, the easier it is to apply the AI technology. The difficulty of the requirements can be assessed, as depicted in Figure 3, by considering the surveillance assets currently in operation, amount of information collected, and tasks to be analyzed.

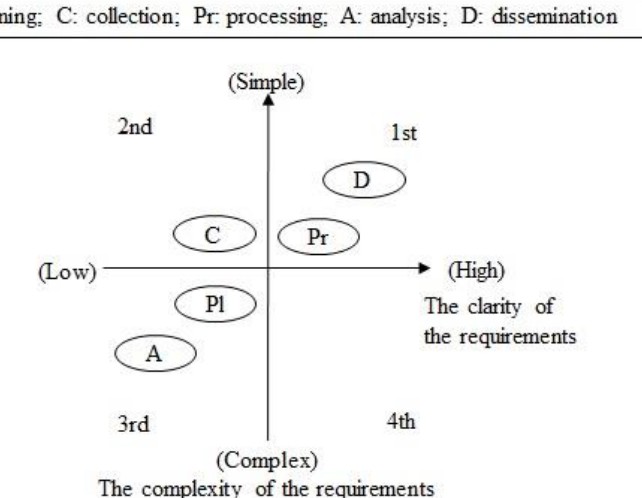

**Figure 3.** Difficulty of the requirements.

During the planning and collection stages, MS Excel was assessed to be sufficient for the operational planning and collection of currently operational surveillance and reconnaissance assets without the application of any AI technology. However, the clarity of the requirements was observed to be low to apply AI technology to the planning and collection of future surveillance and reconnaissance assets because the characteristics of the surveillance and reconnaissance assets and sensors to be acquired in the future are still unknown. According to the assessment results, the planning stage was in the third quadrant of graph; thus, $R_{planning}$ was allocated 0 points. The collection stage was in the second quadrant of the graph; thus, $R_{collection}$ was allocated 1 point.

The requirements for single-source analysis during the processing stage are clearer and simpler than those for all-source analysis during the analysis stage. Further, the requirements during the dissemination stages exhibit high clarity and low complexity. In addition, $R_{processing}$ and $R_{dissemination}$ were allocated 2 points because the processing and dissemination stage were in the first quadrant of the graph. $R_{analysis}$ received 0 points because the analysis stage was in the third quadrant. In terms of difficulty, the requirements of the processing and dissemination stage were assigned a high priority with 2 points because they were clear and simple.

In Step 2, we assessed the readiness of the data, as depicted in Figure 4, corresponding to each domain of the military intelligence service. First, this required the detection of potential data to be applied to each stage by the AI technology and the determination of whether the format was machine-readable. In the data readiness graph, the X-axis represents the availability of data, with availability increasing

towards the right. The Y-axis represents the machine readability of the data, with the level of machine readability increasing in the vertical direction.

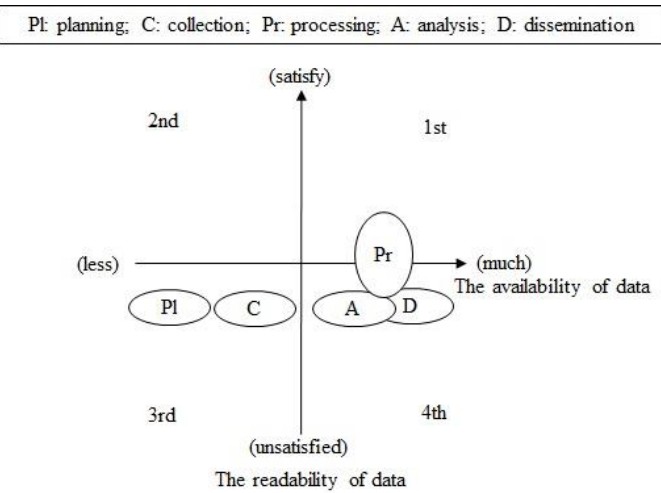

**Figure 4.** Data readiness.

The planning and collection stages are affected the most significantly by the acquisition of future surveillance and reconnaissance assets. The AI algorithm to be applied depends on the characteristics of the additional assets and sensors. Therefore, during the planning and collection stages, the availability of the data is currently low, and the format of the collected data cannot be confirmed; therefore, the data readiness was also assessed as being in the third quadrant of the graph, with $D_{processing}$ and $D_{collection}$ allocated 0 points each.

By contrast, during the processing, analysis, and dissemination stages, military intelligence data accumulated over the past 10 years are retained, and the most of the data are converted into images, Hangul files, and PowerPoint files, thereby satisfying the second level criteria specified in Table 3. In particular, certain data from the processing stage comply with the third level requirements in Table 3, and can be converted into the XML format for processing. Thus, the processing stages spanned the first and fourth quadrants of the data readiness graph; thus, $D_{processing}$ was allocated 1.5 points, which was calculated as (2 + 1)/2. The analysis and dissemination stages were assessed to lie within the fourth quadrant; hence, $D_{analysis}$ and $D_{dissemination}$ were allocated 1 point each. In terms of data readiness, the processing stage had the highest priority with 2 points because a large amount of data was suitable for processing by AI technology through machine reading.

The confidentiality of the data depends on the type of the information and not on the stage of information circulation. For example, secrecy increased along the following hierarchy: "open-source intelligence < imagery intelligence < signal/human intelligence". Notably, the level of confidentiality is proportional to the difficulty of sharing and disclosing data. Thus, the confidentiality of the data is an additional consideration.

In Step 3, AI technology level is evaluated. Table 4 shows the technologies and maturity levels that can be applied to each domain of military intelligence areas. Machine learning and reasoning techniques can be applied to the planning stage, as the optimal plan should be formulated according to the mission, characteristics of surveillance and reconnaissance assets and sensors, weather conditions, among others. The collection stage can be applied via machine learning, recognition, reasoning, and application technologies to select the information collected using the sensors of surveillance and reconnaissance assets that show signs of threat and transfer information to the processing stage.

**Table 4.** Applicable technology and maturity level for military intelligence areas.

| Stage | Work | Applicable Technology (Maturity) |
|---|---|---|
| Planning | Establish a collection plan while considering mission, sensors, and weather, among others | Machine Learning/Inference (3.5) |
| Collection | Identify the data to be processed Capture threat signs | Machine Learning/ Inference (3.5) Recognition intelligence (3.5) Application (2) |
| Processing | Process information such as imagery, signals, and open- source. | Machine Learning (3.5) Recognition intelligence (3.5) |
| Analysis | All-source intelligence analysis Threat signs/time-series target analysis | Machine Learning/Inference (3.5) Application (2) |
| Dissemination | Inquiry of the target information | Visualization (5) Pub/sub (5) Machine Learning/Inference (3.5) |

In the processing stage, machine learning and cognitive intelligence can be applied to information such as images, signals, and open-source data. In the analysis stage, machine learning, inference, and application technology can be used to analyze the threats due to synthesizing the intelligence processed by all the sources or to perform time-series target analyses from the past to present. Finally, the dissemination stage can apply not only the UI/UX visualization technology, but also machine learning and reasoning that analyzes user-usage patterns and recommends relevant information.

The maturity level of the technology applied to the military intelligence tasks was inspired by the study conducted by Oh Jin-tae [56]. The aforementioned study conducted a questionnaire survey among AI experts regarding maturity levels in technology hype cycles corresponding to 13 AI sub-technologies. The results of the positioning analysis for the AI sub-technologies have been depicted by mapping them to the AI technology classification presented in Table 1. Machine learning and inference applied during planning stage were marked to correspond to 3.5, which is a level intermediate to the third (bubbles removal) and fourth stages (re-lighting).

The applications employed in the collection and analysis stages were assigned scores of 2, and machine learning and inference were assigned scores of 3.5. Further, machine learning and cognitive intelligence applied during the processing stage were assigned scores of 3.5. During the dissemination stage, it is appropriate to apply visualization technology, and when AI technology is applied, it is given a score of 3.5 through machine learning and inference. The maturity level of the technology is normalized to allow it to be combined with other assessment criteria.

In Step 4, the effect of technology application is predicted. The effect of the application of the core technologies applicable to intelligence areas was predicted, as presented in Table 5, by referring to the automation value of the technologies used in Project Maven [50]. The effect of the technologies applicable to military intelligence areas.

In the processing stage, computer-vision and cognitive-intelligence technologies that can be applied to the images, signals, and open-source information were predicted to serve as an automation index of 50 times.

Figure 5 shows the maturity of the core technologies to be applied to each detailed military intelligence area mentioned in Table 5 and the result of the technology application. In the graph depicting the technology maturity and application effect, the X-axis represents the maturity of the technology, with the maturity level increasing toward the right. The Y-axis represents the effect of applying the technique, with the effect increasing in the vertical direction.

**Table 5.** Applicable technology and application effect by military intelligence areas.

| Stage | Work | Applicable Technology (Maturity) | Technology Application Effect |
|---|---|---|---|
| Planning | Establish a collection plan while considering mission, sensors, and weather, among others | Machine Learning/Inference (3.5) | Sensor mgt./ optimization: 5× |
| Collection | Identify the data to be processed Capture threat signs | Machine Learning/Inference (3.5) Recognition intelligence (3.5) Application (2) | Optimization: 5× |
| Processing | Process information such as imagery, signals, and open- source. | Machine Learning (3.5) Recognition intelligence (3.5) | Computer vision/AI: 50× Voice-text conversion: 3× |
| Analysis | All-source intelligence analysis Threat signs/time-series target analysis | Machine Learning/Inference (3.5) Application (2) | Information fusion: 5× |
| Dissemination | Inquiry of the target information | Visualization (5) Pub/sub (5) Machine Learning/ Inference (3.5) | UI/UX: 2× Transfer data: 1× |

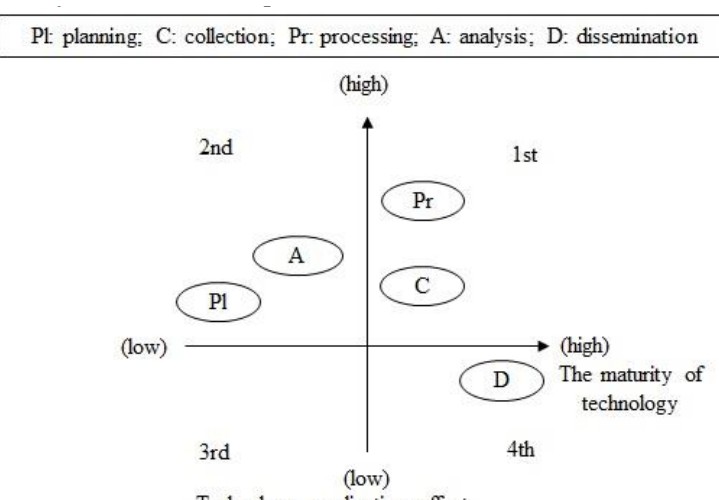

**Figure 5.** Technology maturity and application effect.

In terms of the maturity of AI technology, the inference and application technologies applied during the planning and analysis stages were located on the left side of the graph because of their low maturity, and machine learning and cognitive intelligence applied during the processing and collection stages were located on the right side of the graph because of their high maturity levels.

The application effects of the AI technologies were predicted to be similar in most stages, but it was predicted that applying computer vision/AI to image information during the processing stage would result in 50-fold improvement and would be assessed as the highest. In terms of the maturity of the core technologies and the effects of applying the technologies, the processing stage was accorded a high priority and allocated 2 points.

### 4.3. Results of Priority Determination

Table 6 depicts the results of priority determination obtained by applying the AI technology during the planning, collection, processing, analysis, and dissemination stages, which are the constituent domains in military intelligence.

$$
\begin{aligned}
P_{processing} &= R_{processing} + D_{processing} + TM_{processing} + TE_{processing} \\
&= 2 + 1.5 + 1.4 + 2 \\
&= 6.9
\end{aligned}
\tag{2}
$$

**Table 6.** Priority-determination results obtained upon applying the AI technology to the military intelligence areas.

| Stage / Attribute | Requirement | Data | Technology Maturity | Technical Effect | Total |
|---|---|---|---|---|---|
| Planning | 0 | 0 | 1.4 | 1 | 2.4 |
| Collection | 1 | 0 | 0.8 | 1 | 3.8 |
| Processing | 2 | 1.5 | 1.4 | 2 | 6.9 |
| Analysis | 0 | 1 | 0.8 | 1 | 2.8 |
| Dissemination | 2 | 1 | 0.8 | 1 | 4.8 |

As a result of applying the priority calculation formula, the processing stage was found to have the highest priority at 6.9 points as shown in Equation (2). In terms of the difficulty of the requirements, the requirements in the processing stage were simpler and clearer because of the execution of a source-specific analysis based on currently collected information. The processing stage was located in the first quadrant of the requirement difficulty graph depicted in Figure 3, with $R_{processing}$ allocated 2 points.

In the case of data readiness, the AI technology could be applied immediately as the information collected in the past was readily available (availability). In addition, the information stored in the database corresponded to the second and third stages, and it was machine readable (readability). The processing stage was located across the first and fourth quadrants of the data readiness graph presented in Figure 4, and $D_{processing}$ was allocated 1.5 points as the median score.

Furthermore, machine learning and cognitive functions could be applied to information processing, and the maturity of this technology was ascertained to be between the third (bubble-removal) and the fourth stages (re-lighting). The effect of applying cognitive functions was observed to be the most pronounced in terms of technology-application effects. The processing stage was awarded 1.4 points at the technological maturity ($Tm_{processing}$) level, and 2 points were allocated, in the first quadrant of the technical application effects ($Te_{processing}$) graph in Figure 5. The priority of applying AI technology to the military intelligence area was processing > dissemination > collection > analysis > planning.

The above example is used by policy departments for decision-making purposes. When system development departments assess the priorities by placing higher weights on the development elements, the priorities may vary, as indicated in Table 7.

**Table 7.** Priority-determination result for system development department.

| Stage / Attribute | Requirement | Data | Technology Maturity/Effect | Technical Effect | Total |
|---|---|---|---|---|---|
| Planning | 0 | 0 | 2.8 | 1 | 3.8 |
| Collection | 1 | 0 | 1.6 | 2 | 4.6 |
| Processing | 2 | 4.5 | 2.8 | 2 | 11.3 |
| Analysis | 0 | 3 | 1.6 | 1 | 5.6 |
| Dissemination | 2 | 3 | 1.6 | 1 | 7.6 |

Weight is allocated to evaluation factors related to development: $C_d = 3$ and $C_{Tm} = 2$.

If three times the weight ($C_d = 3$) is assigned to the data readiness level and twice the weight ($C_{Tm} = 2$) is allocated to the maturity level of the technology, the results presented in Table 7 can be obtained. From the perspective of the system development department, the priority can be calculated in the order of processing > dissection > analysis > collection > planning.

## 5. Conclusions

In this study, we suggested a method of determining the priority regarding on which domain of military intelligence areas should be applied by the AI technology. The priority-determination method comprised the process of selecting the technology application domain, presenting the criteria

for prioritization, and determining the priority. The priority of the application of the AI technology was determined by considering the difficulty assessment of requirements, readiness of data, maturity of the AI technology, and technical-use effect according to the criteria for prioritization.

The military intelligence service comprised the following five stages of information circulation: planning, collection, processing, analysis, and dissemination. Upon determining the priority for applying the AI technology to the military intelligence areas the priority for the "processing stage" was the highest. The processing stage was rated higher than other stages in terms of data availability and readability, and it had clear and simple requirements. AI technology, such as machine learning and cognitive functions that could be applied to the processing stage, was on the middle level of the third stage (bubble removal) and fourth stage (re-lighting), and the effect obtained upon applying the technology was also evaluated to be the highest.

The military intelligence service is composed of the following five stages of information circulation: planning, collection, processing, analysis, and dissemination. Upon determining the priority of applying AI technology to the respective areas of military intelligence, the priority for the "processing stage" was found to be the highest. This stage was rated higher than the other stages in terms of data availability and readability, and it had clear and simple requirements. AI technology, such as machine learning and cognitive functions that could be applied to the processing stage, was at the middle level of the third (bubble removal) and fourth (re-lighting) stages, and the effect obtained upon applying the technology was also evaluated to be the highest. Regarding the feasibility of the method proposed in this study, the ROK MND is working to incorporate a suitable direction for applying the AI technology in the military intelligence field into the "National Defense Intelligence Development Policy" based on the research results.

This study was conducted for military intelligence services, and, in the future, it will be necessary to expand the research scope to surveillance and reconnaissance weapon systems. Future surveillance and reconnaissance weapon systems are expected to be of various forms, implemented using small drones, autonomous driving, IoT, and AI technologies. As a result, cyber threats are expected to increase owing to the frailties of advanced technologies. Thus, the introduction of advanced surveillance and reconnaissance weapon systems will require adequate methods for assessing cyber threats [57]. This will necessitate research on the cyber threat assessment of surveillance and reconnaissance weapon systems in the near future.

The proposed priority-determination method for the use of the AI technology in this study suggest ways to efficiently improve work and effectively execute the budget in response to the changes in the defense environment, such as changes in the weight of work due to the rapid increase in military intelligence, reduction in the number of troops, and efficient execution of limited defense budget. This is a method that considers both business-related and technical characteristics, such as requirements and technical maturity and application effects. It can be applied to other defense services in addition to the military intelligence services.

**Author Contributions:** Writing-Original draft preparation, S.C.; Writing-Review and editing, W.S., N.K., and J.J.; Supervision, H.P.I. All authors have read and agreed to the published version of the manuscript.

**Funding:** This research was supported by Institute of Information & communications Technology Planning & Evaluation (IITP) grant funded by the Korea government (MSIT) (No. 2019-0-00099, Formal Specification of Smart Contract).

**Conflicts of Interest:** The authors declare no conflict of interest.

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
