# Peer review of "Priority Determination to Apply Artificial Intelligence Technology in Military Intelligence Areas"

_electronics, doi:10.3390/electronics9122187_

Round 1

Reviewer 1 Report

In this paper, the authors summarize the concept and work status of military intelligence, and the development direction of military information systems. Moreover, a priority determination method is proposed for military intelligence. Finally, the authors validate the outperformance of the method proposed. AI technology can be utilized to visualize the battlefield situation and carry out other military information analysis. The topic of this paper is interesting, but reviewer has the following questions needed further consideration.

  1. The authors should revise the abstract, because the contribution of the paper is not persuasive enough.
  2. In Chapter 2.3, it is a suggestion that the authors should provide some real military examples of the National strategy.
  3. There are some typos in Fig. 2, and the authors should explain what is the specific of 5 researchers and 19 military officers.
  4. 3 is confusing and is in need of the scientific evidence to substantiate the claims about the difficulty of the requirements. Moreover, Fig. 4 and Fig. 5 have the same problems.
  5. In Chapter 4, the authors should explain the clarity of the requirements clearly, e.g. what is its relationship between intelligence technology?
  6. The paper fails to verify the validity of the method, and the simulation for proposed method should be carried out to show the efficiency for military intelligence service.

Some state-of-the-art works about advanced ML methods for compelling wireless applications, such as ‘Thirty Years of Machine Learning: The Road to Pareto-Optimal Wireless Networks’, just to name one.

Reviewer 2 Report

Overall, this manuscript is fairly written. The introduction section significantly racks the discussions and reviews of the current practices and limitations related to military AI work. The authors also need to discuss why the proposed work is important in harnessing AI in military intelligence systems. Also, the authors may need to provide specific examples in detail and application fields that can benefit from AI in military services. The proposed methods and application domain in the manuscript is too general, missing many supporting details. For example, when discussing the size and readiness of data, the authors must provide tangible examples with the size of data and variability in data readiness in the field of military AI. Most parts of this manuscript provide very general reviews on machine learning technology. The authors need to focus more on what specific military-related applications can potentially benefit from machine vision.

This journal aims to publish experimental and theoretical results in as much detail as possible. Overall, this manuscript fails to meet the scope of this journal since the authors did not provide a detailed description of their methods, specific application domains, and importantly missing tangible examples and results.

Reviewer 3 Report

The article is written in a clear and transparent manner for the reader. The assumptions and goals were presented in a simple way at the beginning of the text. The introduction briefly introduces the reader to the issue considered in the text of the article.

Proposals for changes, questions, and comments:
Table 1 (line 159). It is proposed to add a column that identifies the area of ​​an artificial intelligence operation, to which a specific type is assigned in the Type column.

In the case of Figure 1 (line 171) and step 2, data quality is an important factor influencing the applicability of AI methods. Many AI projects, despite having a large amount of data, suffer from too low quality of this data. In addition to data availability, I suggest adding a data quality condition as an important factor.

Line 287. I have to find out. Do you mean MS Excel?

To sum up, the text presented by the authors, the research methodology, and conclusions allow evaluating the article positively.

The question I have after reading the article is related to the nature of the science journal Electronics and the nature of the article. In my opinion, the text deals more with issues related to military development management than broadly understood research in the field of electronics. Please explain why your article is to be published in this journal.

Reviewer 4 Report

Interesting article, I am recommending accept, but I have a few comments/recommendations that you should consider for improving this study.

Comment 1: You should compare similar national frameworks and relevant technologies, e.g.  Japan has a very strong AI national framework, including an artificial brain, while Boston Dynamics has some interesting robots, which are not mentioned in this text. I understand that AI and robotics is not always related, but in a military context, they seem inseparable. I am not saying that you should rewrite the entire paper, but I suggest that you mention such national frameworks/ technology in a short paragraph, probably in a discussion section before the conclusion, or alternatively, in the introduction section. 

Comment 2: there is a lot of work on standardisation of cyber risk in AI, IoT and CPS, among other new and disruptive technologies. You should consider including a short discussion - few sentences - on how these topics/studeis are related to your research, see e.g.:  AI and IoT risk: https://doi.org/10.1186/s42400-020-00052-8 and standardisation of cyber risk: https://doi.org/10.1007/s42452-019-1931-0

Comment 3: you should include a short text after figure 5 (and after each figure) explaining the findings of the figure. At present, it just feels like its being placed there out of context.

Comment 4: The same comment applies for Tables. e.g. Table 7 feels very relevant, but there is no text after the table, you should include at least one sentence explaining the contribution (meaning) of table 7 - and all other tables. This is just one sentence, which you should be able to write relatively easy, because you have created the table, but at present, its left for the reader to make sense and try to understand the meaning/value of the data in the table. 

Comment 5: section 3 discusses the method for priority determination in military intelligence. this section could benefit from a short review of cyber risk, it seems that the word 'risk' is completely missing from this entire text. You should really think about the risks (cyber) when studying and discussing AI in the military context. If you refer back to my comment 2, addressing that comment, could also address my comment 5. Its just a bit strange not to have any discussion on cyber risks in such article. Although I am recommending accept, based on the quality of this text, I am also strongly suggesting that you consider the comment 2 and comment 5 in your revisions. Good luck. 

Round 2

Reviewer 2 Report

After the revision, the authors provide details on their approaches. The revised version includes simple equations, tables with results, and discussions to justify their approach. This reviewer provides minor comments to finalize the manuscript, which follows:

  1. Typo in Line 227, 235, 317, 473, and 547
  2. Lines 242-48 and Table 3: The notation must be consistent. The authors use “Stage” in the text and “Level” in the table. “Son” in Table 3 may be “JSON”
  3. There are mixed uses of “step X” and “Step X.” The authors must stick to one notation.
  4. No numbering for equations.
  5. Line 467, What is a Hangul file?
